# Resistance in Lung Cancer Immunotherapy and How to Overcome It: Insights from the Genetics Perspective and Combination Therapies Approach

**DOI:** 10.3390/cells14080587

**Published:** 2025-04-12

**Authors:** Paweł Zieliński, Maria Stępień, Hanna Chowaniec, Kateryna Kalyta, Joanna Czerniak, Martyna Borowczyk, Ewa Dwojak, Magdalena Mroczek, Grzegorz Dworacki, Antonina Ślubowska, Hanna Markiewicz, Rafał Ałtyn, Paula Dobosz

**Affiliations:** 1Chair of Pathomorphology and Clinical Immunology, Poznan University of Medical Sciences, 61-701 Poznan, Poland; hanna.chowaniec@gmail.com (H.C.); jczerniak@ump.edu.pl (J.C.); dwojakewa@gmail.com (E.D.); gdwrck@gmail.com (G.D.); paula.dobosz@gmail.com (P.D.); 2Université Paris-Saclay, UVSQ, INSERM, END-ICAP, 94805 Versailles, France; mmaria.stepien@gmail.com; 3Doctoral School, Medical University of Lublin, 20-954 Lublin, Poland; 4Faculty of Biology, University of Basel, 4123 Basel, Switzerland; kalitakata05@gmail.com; 5Department of Endocrinology, Internal Medicine and Metabolism, Poznan University of Medical Sciences, 61-701 Poznan, Poland; martyna.borowczyk@gmail.com; 6Department of Pathomorphology, University Clinical Hospital, 61-701 Poznan, Poland; 7Department of Neurology, University Hospital Basel, 4123 Basel, Switzerland; m.mroczek888@gmail.com; 8Department of Biostatistics and Research Methodology, Faculty of Medicine, Collegium Medicum, Cardinal Stefan Wyszynski University of Warsaw, 02-004 Warsaw, Poland; a.slubowska@uksw.edu.pl; 9Department of Histology and Embryology, Faculty of Medicine, Medical University of Warsaw, 02-004 Warsaw, Poland; 10Department of Methodology, Faculty of Medicine, Medical University of Warsaw, 02-004 Warsaw, Poland; 11IT Department, Poznan University of Medical Sciences, 61-701 Poznan, Poland; rafaltyn@gmail.com

**Keywords:** lung cancer, tumour microenvironment, checkpoint inhibitors, immunotherapy resistance

## Abstract

Lung cancer with the highest number of new cases diagnosed in Europe and in Poland, remains an example of malignancy with a very poor prognosis despite the recent progress in medicine. Different treatment strategies are now available for cancer therapy based on its type, molecular subtype and other factors including overall health, the stage of disease and cancer molecular profile. Immunotherapy is emerging as a potential addition to surgery, chemotherapy, radiotherapy or other targeted therapies, but also considered a mainstay therapy mode. This combination is an area of active investigation in order to enhance efficacy and overcome resistance. Due to the complexity and dynamic of cancer’s ecosystem, novel therapeutic targets and strategies need continued research into the cellular and molecular mechanisms within the tumour microenvironment. From the genetic point of view, several signatures ranging from a few mutated genes to hundreds of them have been identified and associated with therapy resistance and metastatic potential. ML techniques and AI can enhance the predictive potential of genetic signatures and model the prognosis. Here, we present the overview of already existing treatment approaches, the current findings of key aspects of immunotherapy, such as immune checkpoint inhibitors (ICIs), existing molecular biomarkers like PD-L1 expression, tumour mutation burden, immunoscore, and neoantigens, as well as their roles as predictive markers for treatment response and resistance.

## 1. Introduction

Lung cancer remains one of the most challenging malignancies worldwide, with particularly concerning statistics in Europe. Despite recent medical advances, mortality rates closely mirror incidence rates, with lung cancer deaths among women now exceeding breast cancer mortality [1]. In Europe, the annual incidence reaches 350,000 new cases in men (60,000 per 100,000) and 300,000 in women (50,000 per 100,000), with especially high rates in Eastern Europe, including Poland [2]. The percentage of different types of lung cancers is presented in Figure 1. While male incidence has begun to decline due to decreased smoking rates, the persistently high number of female smokers presents an ongoing concern, contributing to approximately 250,000 female and 300,000 male deaths annually in Europe [2].

In recent years, immunotherapy has revolutionised the treatment landscape for lung cancer, particularly through immune checkpoint inhibitors (ICIs). These agents work by stimulating the immune system to attack cancer cells, primarily through triggering and activating tumour-specific T cells [3]. The effectiveness of this approach relies on inducing or restoring the immune system’s capacity, especially T cells, to recognise and eliminate cancer cells [3]. This process involves a delicate balance known as cancer immunoediting, where the immune system both protects against tumour development and inadvertently promotes the survival of those cancer cells that escaped from immunosurveillance [3,4,5,6].

The success of immunotherapy, particularly PD-1/PD-L1 blockade, has been most pronounced in genetically complex tumours with high mutational loads, such as non-small cell lung cancer (NSCLC) [3,7]. Recent clinical trials have demonstrated significant advances, with the IMpower010 trial showing improved disease-free survival with adjuvant atezolizumab in resected NSCLC, and the CheckMate 816 trial establishing the benefits of neoadjuvant nivolumab combined with chemotherapy [8]. For small cell lung cancer (SCLC), the integration of ICIs into first-line chemotherapy regimens has become standard practice, with promising results from trials like ADRIATIC [9].

However, response rates to checkpoint inhibitors vary considerably, typically ranging from 8% to 30% [3,10,11,12,13]. While the durability of response is impressive—with 70–89% of responding patients maintaining remission for 6–14 months—these statistics highlight a critical challenge: many patients either fail to respond initially or develop resistance over time [3,10,11,12,13]. This variable response pattern occurs despite lung cancer’s relatively high mutation load, which theoretically provides numerous epitopes for cytotoxic T-cell targeting [14].

Furthermore, checkpoint inhibition can trigger a range of immune-related adverse events (irAEs) [13,15]. While most common side effects such as fatigue, skin rashes, and loss of appetite are manageable, more severe complications including autoimmune responses, pneumonitis, and even fatal pulmonary toxicity can occur in approximately 3% of anti-PD1-treated patients [15,16,17,18,19]. In extreme cases, immunotherapy may paradoxically induce hyperprogression of cancer, leading to rapid clinical deterioration [18,20,21,22].

Environmental factors, particularly tobacco smoking, significantly influence therapy success. Recent analyses have revealed correlations between smoking history and the expression of crucial immune-related genes, including the poliovirus receptor (PVR) and methylation patterns of PD-L1 and ICOSL genes [23,24,25]. These findings underscore the complex interplay between environmental factors and immune response in cancer treatment. To address these challenges, researchers are exploring various strategies, including novel checkpoint targets such as LAG-3, TIM-3, and OX-40 [26], combination therapies, and innovative delivery methods utilising nanomedicine [27]. Understanding and overcoming resistance mechanisms, particularly those associated with genetic mutations like STK11, KEAP1, and JAK1/2, has become crucial for improving patient outcomes [28].

In summary, the benefits of immunotherapy have changed the approach to treatment, especially in patients with NSCLC and SCLC. Patients who respond to treatment can experience remission for 6–14 months or longer [29]. Particularly promising results are observed in tumours with a high mutational burden, where immunotherapy can more effectively stimulate the immune system [30]. Groundbreaking studies such as Impower010 and Checkmate 816 have confirmed the efficacy of adjuvant and neoadjuvant therapy based on PD-1/PD-L1 inhibitors [31]. New strategies are being developed, including nanomedicine and combination therapies, that may increase the efficacy of immunotherapy [32]. Variable efficacy of immunotherapy—responses to ICIs range from 8% to 30%, meaning that many patients do not benefit from this therapy [33]. Some patients, despite an initial response, develop resistance to treatment, which limits the long-term efficacy of therapy. Smoking affects the expression of genes associated with the immune response (e.g., PVR, PD-L1, ICOSL), but the mechanisms of this impact require further study [34]. As an adverse event, immunotherapy can cause serious adverse events, such as autoimmune pneumonia or tumour hyperprogression, but the exact mechanisms of these complications are not yet fully understood. We need new therapeutic targets to identify alternative checkpoints (e.g., LAG-3, TIM-3, OX-40) and develop effective combination therapies [35].

This review focuses on the critical challenge of immunotherapy resistance in lung cancer, examining its mechanisms from a genetic perspective and exploring promising combination therapy approaches to overcome these limitations.

## 2. Treatment Landscape for Lung Cancer: Immunotherapy in Context

Treatment options for lung cancer include surgery, chemotherapy, radiotherapy, targeted therapies, immunotherapy, and various combinations of these approaches. In most lung cancer cases, immunotherapy has become the leading systemic treatment approach, second in importance only to surgery. This therapy works by leveraging our immune system’s fundamental ability to distinguish between the body’s own cells and foreign entities. Checkpoint inhibitor drugs, which counteract the mechanisms that cancer uses to evade immune detection, have shown exceptional promise in treatment. For this reason, our paper will concentrate primarily on immunotherapy as a lung cancer treatment strategy, while also describing how other therapeutic approaches can complement and enhance immunotherapy outcomes.

### 2.1. Immunotherapy

Immunotherapy, particularly checkpoint blockade therapies, has revolutionised lung cancer treatment by triggering and activating tumour-specific T cells [3]. This approach works through induction or restoration of the immune system’s capacity to recognise and eliminate cancer cells [3], operating within the framework of cancer immunoediting—a fragile balance where the immune system both protects against tumour development while inadvertently promoting the growth of those were able to escape from immunosurveillance [3,4,5,6]. Checkpoint inhibitors have shown particular efficacy in genetically complex tumours with high mutational loads and numerous neoantigens, such as NSCLC [3,7], with proven effectiveness in multiple cancer types including melanoma, head and neck cancer, and metastatic bladder cancer [10]. Despite their promise, response rates vary considerably from 8% to nearly 30%, though responding patients often maintain durable remission for 6–14 months [3,10,11,12,13]. This variable efficacy, despite lung cancer’s relatively high mutation burden [14], underscores our incomplete understanding of the immune synapse between tumour and immune cells. Immune-related adverse events remain a challenge, ranging from common side effects (fatigue, skin rashes) to serious complications like autoimmune responses and pneumonitis, with approximately 3% of anti-PD1-treated patients developing severe pulmonary toxicity [15,16,17,18,19]. Environmental factors, particularly tobacco smoking, significantly influence treatment outcomes by affecting the expression of immune-related genes such as PVR and methylation patterns of PD-L1 and ICOSL genes, highlighting the complex interplay between environmental exposures and immune response in cancer treatment [23,24,25].

### 2.2. Surgery

Surgery remains the primary intervention for early-stage disease (stages I-IIIa), particularly in NSCLC, offering the best chance for cure by physically removing cancerous tissue [36,37,38,39,40,41]. It eliminates immunosuppressive factors such as tumour-associated fibroblasts and extracellular matrix components while releasing tumour-specific antigens that trigger stronger T-cell responses [36,38]. The resulting inflammatory response mobilises immune cells to the area, creating favourable conditions for immune-mediated tumour control [38,39]. This post-surgical immune landscape offers a strategic opportunity for adjuvant immunotherapy to target microscopic residual disease with enhanced efficacy.

### 2.3. Chemotherapy

Chemotherapy, once the standard treatment for advanced lung cancer, now demonstrates its greatest value in combination with immunotherapy [42]. This synergistic approach potentiates immunotherapy effects by increasing tumour antigenicity and disrupting the immunosuppressive tumour microenvironment [43]. Pivotal studies have established the clinical benefit of these combinations. Gandhi et al. (2018) demonstrated that adding pembrolizumab to platinum-based chemotherapy with pemetrexed significantly improved overall survival, progression-free survival, and response rates in patients with metastatic non-squamous NSCLC without *EGFR* or *ALK* alterations [44]. Similarly, Socinski et al. (2018) showed that atezolizumab combined with bevacizumab and chemotherapy improved survival outcomes regardless of PD-L1 expression or genetic alteration status [45]. These landmark studies have transformed the first-line treatment paradigm for advanced NSCLC without targetable mutations, establishing chemoimmunotherapy combinations as the standard of care. The immunomodulatory effects of cytotoxic agents appear to enhance immune recognition of tumour cells while simultaneously reducing immunosuppressive elements within the tumour microenvironment, creating conditions more favourable for effective immune surveillance and attack.

### 2.4. Radiotherapy

Radiotherapy, one of the oldest cancer treatment modalities [46], has evolved from a primarily local tumour control strategy to a potential systemic immune modulator when combined with immunotherapy. While traditional radiation effects include direct DNA damage [47,48] and cell cycle disruption, its immunomodulatory effects have gained significant attention. Radiation enhances the recognition of tumour-associated antigens by the immune system, increasing the recruitment and activation of immune cells [49], which can prime the immune system for checkpoint inhibitor therapy. However, radiation’s immune effects are complex, as it can also induce immunosuppression through increased regulatory T cells and myeloid-derived suppressor cells [50].

Particularly relevant to immunotherapy integration is radiation’s ability to modulate PD-L1 expression, directly affecting tumour responsiveness to checkpoint inhibitors [51]. The combination of radiotherapy with immunotherapy has shown promising clinical results, as demonstrated by Theelen et al. (2021), who found improved overall survival in NSCLC patients receiving radiotherapy with pembrolizumab [52]. This approach can potentially induce the abscopal effect, where radiation-enhanced immune responses affect non-irradiated metastatic sites, transforming a local treatment into a systemic therapy. By strategically sequencing radiation with immunotherapy, clinicians can potentially leverage both the direct cytotoxic effects of radiation and its immune-stimulating properties to overcome resistance mechanisms and enhance treatment outcomes.

### 2.5. Targeted Therapies

Targeted therapies have transformed lung cancer treatment by precisely inhibiting specific oncogenic drivers like *EGFR, ALK, ROS1,* and *KRAS G12C* mutations, as indicated in Table 1 and Table 2 below [53,54]. Despite impressive initial responses in patients with actionable mutations, resistance inevitably develops, necessitating alternative strategies [55,56]. The integration of targeted therapies with immunotherapy represents an emerging approach to overcome resistance mechanisms and extend treatment benefits. Studies by Nakagawa et al. (2019) demonstrated that dual blockade of the EGFR and VEGF pathways using EGFR TKIs with ramucirumab (a VEGFR2 antagonist) improved progression-free survival in EGFR-mutated metastatic NSCLC [57].

This combination approach addresses a significant challenge: patients with oncogene-driven tumours typically show lower response rates to immune checkpoint inhibitors as monotherapy, likely due to lower tumour mutational burden and less immunogenic microenvironments [3,7]. By strategically combining targeted agents with immunotherapy, treatment can simultaneously address the oncogenic driver while enhancing immune recognition and attack [58,59]. Emerging evidence suggests that certain targeted therapies may favourably alter the tumour immune microenvironment, potentially sensitising previously resistant tumours to immunotherapy [56,60].

The extensive range of FDA and EMA-approved targeted therapies for specific genetic alterations [59], including newer agents for previously “undruggable” targets like KRAS G12C, provides multiple opportunities for rational combination strategies [55,61]. As research advances, optimising the sequencing, dosing, and selection of specific combinations based on molecular profiles will be crucial for maximising efficacy while managing overlapping toxicities. This precision medicine approach, integrating both targeted therapy and immunotherapy based on comprehensive genetic profiling, represents the frontier of personalised treatment for lung cancer patients [45,53,54,55,56,58,59,60,61,62].

**Table 1 cells-14-00587-t001:** Targeted therapies approved by the FDA and European Medicines Agency (EMA), available in Poland versus selected EU countries, designated for lung cancer treatment—included in the national oncologic treatment schemes [59].

*Drug Name and Availability in Poland Versus in Selected EU Countries*	*Poland*	*Italy*	*Spain*	*Sweden*	*Slovenia*	*Czechia*	*Estonia*
*Osimertinib* *2nd line post EGFR TKI, T790M+*							
*Osimertinib* *1st line, EGFR M+*							
*Crizotinib* *1st line, ALK+ patients*							
*Crizotinib* *any line, ROS+ patients*							
*Alectinib* *1st line, ALK+ patients*							
*Dabrafenib-trametinib* *BRAF+ patients*							

Green = widely available. Yellow = available with restriction. Red = not available.

**Table 2 cells-14-00587-t002:** Selected molecularly targeted therapies approved by the FDA and/or EMA for use in lung cancer or any cancer as long as they possess certain genetic variants (so-called oncoagnostic approach, in line with modern precision medicine), often used in combination therapies. It is important to bear in mind that this table presents only some of the most prominent targeted therapies and mutations. The landscape continues to evolve with ongoing research and clinical trials [45,52,63,64,65,66,67,68,69,70,71,72,73,74,75,76,77,78,79,80,81,82,83,84,85,86,87,88,89,90,91,92,93,94,95,96,97,98,99,100,101,102,103,104,105,106,107,108,109,110,111,112].

Drug	Targeted Mutation	FDA Approved	Available in EU	Stage in Clinical Trials (If Not FDA Approved)
*Gefitinib*	EGFR exon 19 deletion/L858R mutation	Yes	Yes	-
*Erlotinib*	EGFR exon 19 deletion/L858R mutation	Yes	Yes	-
*Osimertinib*	EGFR T790M resistance mutation	Yes	Yes	-
*Alectinib*	ALK rearrangements	Yes	Yes	-
*Brigatinib*	ALK rearrangements	Yes	Yes	-
*Crizotinib*	ALK, ROS1 rearrangements	Yes	Yes	-
*Lorlatinib*	ALK rearrangements	Yes	Yes	-
*Sotorasib*	KRAS G12C mutation	Yes	Yes	-
*Adagrasib*	KRAS G12C mutation	Yes	Yes	-
*Selpercatinib*	RET fusions	Yes	Yes	-
*Pralsetinib*	RET fusions	Yes	Yes	-
*Capmatinib*	MET exon 14 skipping mutation	Yes	Yes	-
*Tepotinib*	MET exon 14 skipping mutation	Yes	Yes	-
*Amivantamab*	EGFR exon 20 insertion mutation	Yes	Yes	-
*Mobocertinib*	EGFR exon 20 insertion mutation	Yes	Yes	-
*Entrectinib*	ROS1, neurotrophic tyrosine receptor kinase (NTRK) gene fusions	Yes	Yes	-
*Larotrectinib*	NTRK gene fusions	Yes	Yes	-
*Nivolumab*	PD-1 (immune checkpoint inhibitor)	Yes	Yes	-
*Pembrolizumab*	PD-1 (immune checkpoint inhibitor)	Yes	Yes	-
*Atezolizumab*	PD-L1 (immune checkpoint inhibitor)	Yes	Yes	-
*Durvalumab*	PD-L1 (immune checkpoint inhibitor)	Yes	Yes	-
*Savolitinib*	MET exon 14 skipping mutation	No	No	Phase 2/3
*Dabrafenib/Trametinib*	BRAF V600E mutation	Yes	Yes	-

### 2.6. Novel Immunotherapeutic Approaches

The evolution of immunotherapy extends beyond PD-1/PD-L1 and CTLA-4 inhibitors to novel checkpoint targets that address resistance mechanisms in lung cancer. Emerging approaches focus on alternative immune checkpoints including LAG-3, TIM-3, and TIGIT [113]. LAG-3 inhibitors like relatlimab combined with nivolumab have demonstrated promising activity in PD-L1-positive NSCLC patients, improving progression-free survival, while eftilagimod alpha shows potential in enhancing pembrolizumab efficacy by activating antigen-presenting cells [114,115]. TIM-3 targeting agents such as sabatolimab and TSR-022 have shown manageable toxicity profiles with improved outcomes when combined with PD-1 inhibitors, though with limited efficacy as monotherapies [26,116]. TIGIT blockade represents another promising pathway, with tiragolumab significantly enhancing progression-free survival when combined with atezolizumab in PD-L1-positive patients, while domvanalimab with zimberelimab has shown encouraging response rates [117,118]. Therapeutic cancer vaccines [119] and innovative combinations like BRAF and MEK inhibitors for BRAF V600E-mutant NSCLC [112] further expand treatment options. These novel approaches emphasise the importance of combinatorial strategies, as most new checkpoint inhibitors demonstrate modest activity alone but enhanced efficacy when integrated with established immunotherapies, targeted agents, or conventional treatments, providing promising avenues to overcome the complex resistance mechanisms in lung cancer [26,113,114,115,116,117,118,120]. Table 3 summarises the key findings from early-phase clinical trials investigating checkpoint inhibitors targeting LAG-3, TIM-3, TIGIT, and ICOS pathways.

## 3. Problem of Resistance to Immunotherapy in Lung Cancer

Immunotherapy, particularly immune checkpoint blockade, has revolutionised cancer treatment, including non-small cell lung cancer. However, resistance to immune checkpoint blockade remains a significant challenge, limiting its therapeutic potential for many patients [121]. Understanding these resistance patterns and their underlying mechanisms is crucial for developing effective treatment strategies.

### 3.1. Types of Resistance

Primary resistance refers to the lack of initial response to immunotherapy, occurring in 40–44% of NSCLC patients treated with immune checkpoint inhibitors as second-line therapy and 21–27% as first-line therapy [122]. The incidence is notably lower (around 10%) when ICIs are combined with chemotherapy as first-line treatment [44,123]. Primary resistance may result from various factors, including low tumour immunogenicity, impaired antigen presentation, and an immunosuppressive tumour microenvironment (TME) [124,125].

Secondary (acquired) resistance develops in patients who initially respond to immunotherapy but later experience disease progression [121]. This affects a significant number of NSCLC patients, with studies reporting secondary resistance rates of 52–57% for first-line and 32–64% for second-line ICI treatment [126]. The mechanisms underlying secondary resistance are complex and may involve changes in tumour cells, the TME, or both [127]. Key mechanisms of resistance include tumour-intrinsic factors such as loss of tumour antigens and neoantigen expression, alterations in antigen presentation machinery, activation of oncogenic signalling pathways, impaired interferon (IFN)-γ signalling, and upregulation of alternative immune checkpoints [128,129].

### 3.2. Mechanisms of Resistance

The mechanisms of resistance to immunotherapy in lung cancer are multifaceted, encompassing genetic alterations, metabolic changes, and environmental factors. Among these, specific genetic mutations play a particularly significant role in determining treatment outcomes.

#### 3.2.1. Genomic Alterations and Associated Factors

Genetic alterations in lung cancer not only affect direct immune responses but also influence metabolic, epigenetic, and microbiome-mediated mechanisms of resistance.

★
***STK11 (LKB1)* Mutations**


*STK11* mutations represent a major challenge in immunotherapy resistance, associated with a non-T cell-inflamed TME and frequently co-occurring with *KRAS* mutations [130,131]. These mutations significantly impact cellular metabolism, leading to increased lactate production and secretion via the MCT4 transporter, which subsequently suppresses antitumour immunity by promoting M2 macrophage polarization and T-cell dysfunction [132]. The impact of *STK11* mutations extends beyond direct metabolic effects, as they also influence epigenetic regulation. DNA methylation and histone modifications can further suppress the expression of genes involved in immune activation, exacerbating resistance to ICIs. This complex interaction suggests potential therapeutic opportunities through DNA methyltransferase inhibitors (DNMTis) or histone deacetylase inhibitors (HDACis) [132].

★
***KEAP1* Mutations**


*KEAP1* mutations present another significant challenge, resulting in hyperactivation of the NRF2 pathway and leading to enhanced oxidative stress resistance and immune evasion [133]. These mutations profoundly affect cellular metabolism, increasing the consumption of glucose and glutamine by tumour cells, thereby depriving effector T cells of essential nutrients and promoting immunosuppression. Recent research has shown promising results in combining ICIs with glutaminase inhibitors to reverse these immunosuppressive effects [134,135,136]. The impact of *KEAP1* mutations is further modulated by epigenetic regulation of the NRF2 pathway and its downstream targets, suggesting that targeting these epigenetic mechanisms might enhance immunotherapy efficacy [133].

★
***JAK1/2* Mutations**


*JAK1/2* mutations disrupt interferon-gamma signalling, significantly impairing antigen presentation and T cell activation [130]. This disruption leads to metabolic dysregulation affecting cell surface marker expression and immune surveillance. Current research is exploring strategies to restore or bypass these pathways, such as using agents that enhance antigen presentation or T cell activation, to improve outcomes in *JAK1/2*-mutant tumours [137,138]. These mutations also influence epigenetic regulation, particularly in the downregulation of interferon-stimulated genes, contributing to resistance mechanisms [139].

The above-described mutations have been summarised in Table 4 below.

#### 3.2.2. Microbiome Influence

The gut microbiome plays a crucial role in modulating responses to immunotherapy across all these genetic contexts. Research has shown that specific bacterial species and overall microbiome diversity can significantly impact ICI efficacy. In patients with STK11 mutations, the immunosuppressive TME can be influenced by the gut microbiome, suggesting potential therapeutic approaches through microbiome modulation [140]. Similarly, the efficacy of metabolic interventions in KEAP1-mutant tumours may be modulated by the gut microbiome [136], while microbiome-based interventions might help restore immune function in JAK1/2-mutant cases [141].

#### 3.2.3. Tumour Microenvironment Factors

The tumour microenvironment (TME) of lung cancer represents a complex and dynamic ecosystem that significantly influences immunotherapy outcomes [142]. The complexity of TME is illustrated in Figure 2. Understanding its components and their interactions is crucial for comprehending resistance mechanisms and developing effective therapeutic strategies.

★
**Cancer-Associated Fibroblasts (CAFs)**


Cancer-associated fibroblasts serve as a major component of the TME, playing a significant role in promoting therapy resistance. These cells secrete extracellular matrix (ECM) components and remodelling enzymes that facilitate tumour progression and treatment resistance through ECM modification. Additionally, CAFs release critical signalling molecules, including TGF-β, VEGF, and IL-6, which actively modulate immune responses and angiogenesis [143]. The targeting of these stromal components has emerged as a promising strategy to combat tumour development and resistance to conventional therapies, potentially transforming lung cancer into a more manageable disease [144].

★
**Immune Cell Composition**


The immune landscape within lung cancer presents a diverse array of cells that influence treatment response. This includes tumour-associated macrophages, which differentiate into either proinflammatory M1 (anti-tumour) or immunosuppressive M2 (pro-tumour) phenotypes, T cells, natural killer (NK) cells, and myeloid-derived suppressor cells (MDSCs). The presence of regulatory T cells (Tregs) and exhausted CD8+ T cells within the TME frequently leads to compromised anti-tumour immunity [145]. A “T cell-inflamed” TME, characterised by high CD8+ T cell infiltration, is typically associated with better responses to immune checkpoint inhibitors, while an immunosuppressive environment dominated by regulatory T cells, MDSCs, and M2 macrophages often leads to treatment resistance [146].

★
**Hypoxia and Angiogenesis**


Hypoxia represents a critical feature of the TME that contributes to immunotherapy resistance. The stabilization of HIF-1α under hypoxic conditions regulates genes involved in angiogenesis, metabolism, and survival, promoting VEGF expression and correlating with poor prognosis in lung cancer patients [147]. Endothelial cells play a vital role in this process through the formation of new blood vessels, with VEGF serving as a key mediator. Beyond VEGF, other angiogenic factors including FGF-2 and HIFs contribute to tumour cell proliferation, migration, and invasion [148,149]. Strategies to alleviate hypoxia, such as HIF-1α inhibitors or agents that normalise tumour vasculature, may enhance the effectiveness of immunotherapy by improving immune cell infiltration and activity [150].

★
**Extracellular Matrix and Signalling Pathways**


The ECM provides both structural support and regulatory functions through biochemical and mechanical signals. In lung cancer, ECM dysregulation promotes tumour cell proliferation, migration, and invasion. Matrix metalloproteinases (MMPs) and other enzymes, including lysyl oxidases and ADAMTS, facilitate metastasis through ECM remodelling [151]. The TGF-β signalling pathway particularly influences immunotherapy resistance by inducing epithelial-mesenchymal transition, enhancing invasion, and suppressing immune responses, contributing to a more aggressive phenotype [152].

#### 3.2.4. Comorbidities Impact

Comorbidities significantly influence the efficacy of immunotherapy, particularly in both NSCLC and SCLC patients. The introduction of immunotherapy, particularly immune checkpoint inhibitors (ICIs), has significantly transformed the therapeutic landscape of lung cancer treatment. ICIs, which primarily function by inhibiting the interaction between PD-1/PD-L1 or CTLA-4 and their respective ligands, have redefined oncological treatment protocols. While immunotherapy offers promising clinical benefits, it is also associated with immune-related adverse events (irAEs), a distinct spectrum of treatment-related complications. Among these, immune-related pneumonitis (IRP) is of particular concern, especially in patients with lung cancer. Although IRP remains a relatively rare occurrence, its potential severity—ranging from mild, manageable symptoms to life-threatening respiratory failure—necessitates careful clinical consideration [153].

Chronic Obstructive Pulmonary Disease (COPD) is a prevalent respiratory condition characterised by persistent symptoms and airflow limitation. It is a heterogeneous disorder encompassing various phenotypes, including emphysema and chronic bronchitis. The overlap between COPD and lung cancer is substantial, with shared risk factors such as smoking and environmental exposures contributing to chronic inflammation and cellular alterations that predispose individuals to oncogenesis. These complexities render COPD a critical factor in the management of lung cancer, particularly in relation to IRP development during ICI therapy. The pathogenesis of COPD involves chronic airway inflammation, structural remodelling, and significant alterations in pulmonary immune responses. These mechanisms may predispose COPD patients to an elevated risk of IRP when treated with ICIs, given their already compromised pulmonary function. A pooled analysis has demonstrated that lung cancer patients with concomitant COPD exhibit a heightened susceptibility to IRP during immunotherapy. Fangyuan presents compelling evidence indicating a significantly elevated risk of immune-related pneumonitis (IRP) in patients with concomitant Chronic Obstructive Pulmonary Disease (COPD). IRP is a potentially life-threatening complication that can severely impair respiratory function and diminish quality of life, often necessitating the discontinuation of life-prolonging immunotherapy. Among the identified risk factors for IRP, COPD has emerged as a critical contributor. Patients with COPD exhibit a heightened susceptibility to IRP due to pre-existing pulmonary inflammation and an already compromised immune response, which may interact with the immunomodulatory effects of immune checkpoint inhibitors (ICIs) [154]. Zhou’s findings underscore the impact of COPD on the risk of IRP in lung cancer patients undergoing immunotherapy. By identifying this high-risk subgroup, oncologists can more effectively weigh the risks and benefits of immunotherapy, facilitating the development of predictive models for IRP and the implementation of preventive strategies to mitigate its occurrence. The identification of high-risk subgroups among lung cancer patients is crucial for optimising the balance between the risks and benefits of immunotherapy. By recognising such populations, oncologists can facilitate the development of predictive models for immune-related pneumonitis (IRP) and implement preventive strategies to mitigate its occurrence.

Chronic obstructive pulmonary disease (COPD) presents a complex interplay with immunotherapy response. Patients with COPD exhibit increased Th1 differentiation in both pulmonary and tumour tissues, potentially influencing the efficacy of immune checkpoint inhibitors (ICIs) [155]. Additionally, the presence of multiple comorbidities, as assessed by the Charlson Comorbidity Index (CCI), has been linked to poorer progression-free survival in individuals undergoing PD-1 inhibitor therapy [156].

The significant heterogeneity observed in studies investigating this relationship may be attributable to variations in study populations, lung cancer types and stages, COPD severity, and differing immunotherapeutic regimens. The elevated odds ratio (OR) highlights a critical clinical consideration for practitioners managing lung cancer patients with pre-existing COPD, as these individuals face a heightened risk of developing IRP—a potentially severe and treatment-limiting adverse event. This increased susceptibility may stem from the underlying inflammatory state and altered pulmonary immune environment associated with COPD, which could interact with the mechanisms of action of ICIs, leading to an exacerbated inflammatory response in the lungs [154].

Furthermore, lung cancer patients with COPD are at an elevated risk of IRP, particularly in the context of radiation therapy. Notably, this heightened risk persists across varying levels of radiation exposure (< 40% and ≥ 40%), suggesting that COPD functions as an independent risk factor for IRP. When compounded by radiation-induced effects, the likelihood of this adverse event may be further increased.

The literature presents conflicting findings regarding the influence of COPD on lung cancer treatment outcomes. While some studies associate COPD with poorer responses to ICIs, others indicate that in patients with COPD, Th1 cell populations are expanded in both the lung and tumour microenvironments. Notably, COPD has been linked to prolonged progression-free intervals in patients treated with ICIs [155]. These findings underscore the need for further investigation into the immune mediators of COPD and their implications for developing novel therapeutic strategies for non-small cell lung cancer (NSCLC).

The impact of comorbidities is particularly noteworthy in small-cell lung cancer (SCLC) patients. While these individuals often exhibit a high mutational burden due to smoking history, the presence of comorbidities such as renal or hepatic impairment can significantly complicate treatment outcomes. The American Society of Clinical Oncology (ASCO) guidelines specifically highlight that patients with poor performance status or significant comorbidities tend to have inferior outcomes with dual ICI therapy, likely due to the poor prognosis associated with their underlying conditions [157]. This challenge is further compounded by the immunosuppressive phenotype of SCLC, characterised by low PD-L1 expression and poor T-cell infiltration, which adds another layer of complexity to ICI efficacy in these patients [158].

### 3.3. Patient Characteristics Associated with Resistance

Primary resistance patterns are particularly evident in specific patient populations. Non-smokers tend to show higher rates of primary resistance, primarily due to their typically lower tumour mutational burden, which correlates with reduced immunogenicity and poor response to ICIs [159]. Patients with more extensive disease burden also frequently exhibit primary resistance, as the extensive disease can create an immunosuppressive tumour microenvironment that hinders effective immune responses. Similarly, patients with low albumin levels, which may reflect poor overall health and a compromised immune system, often show reduced efficacy of immunotherapy [159]. A crucial factor in primary resistance is poor T-cell infiltration, as tumours lacking sufficient CD8+ T-cell infiltration are less likely to respond to ICIs due to insufficient immune activation [160].

Acquired resistance presents a different set of challenges, developing in patients who initially respond to ICIs but later experience disease progression. This form of resistance often involves complex genomic alterations, with mutations in genes such as STK11, B2M, APC, MTOR, KEAP1, and JAK1/2 being implicated in resistance mechanisms [130]. The tumour microenvironment undergoes significant changes in cases of acquired resistance, characterised by increased infiltration of immunosuppressive cells like myeloid-derived suppressor cells and M2 macrophages, along with decreased neoantigen presentation [160]. Additionally, immunophenotypic changes play a crucial role, with some patients showing reduced numbers of tumour-infiltrating CD8+ T cells and decreased PD-L1 expression, while others may exhibit high CD8+ T cell infiltration but elevated expression of other immune-inhibitory molecules [161].

### 3.4. Hyperprogression

Hyperprogression, a phenomenon characterised by accelerated tumour growth following treatment with immune checkpoint inhibitors (ICIs), is associated with poor prognosis in lung cancer. Emerging evidence suggests that this rapid disease progression is driven by a complex interplay of genetic alterations, metabolic dysregulation, microbial influences, and epigenetic modifications.

#### 3.4.1. Genetic Mutations

Several genetic mutations have been implicated in hyperprogression by fostering an immunosuppressive tumour microenvironment (TME) and promoting resistance to ICIs:*STK11 (LKB1)* mutations contribute to an immunosuppressive TME by increasing neutrophil recruitment and proinflammatory cytokine production, leading to poor responses to ICIs and an increased likelihood of hyperprogression [162].*KEAP1* mutations drive hyperactivation of the NRF2 pathway, which enhances immune evasion and metabolic reprogramming, ultimately supporting tumour growth and ICI resistance [137].*JAK1/2* mutations impair interferon-gamma signalling, disrupting antigen presentation and T-cell activation. This immune dysfunction can facilitate rapid tumour progression during immunotherapy [137].

#### 3.4.2. Metabolic Dysregulation

Metabolic reprogramming in tumours can create an immunosuppressive TME that fosters hyperprogression. For example, increased glycolysis and lactate production, particularly in tumours harbouring *STK11* mutations, suppress antitumour immunity, thereby exacerbating resistance to ICIs [163].

#### 3.4.3. Microbial Influences

The gut microbiome plays a crucial role in modulating systemic immune responses and the efficacy of ICIs. Dysbiosis or specific microbial compositions can alter immune cell function and cytokine production, thereby influencing tumour progression and potentially contributing to hyperprogression [163].

#### 3.4.4. Epigenetic Modifications

Epigenetic alterations, including DNA methylation and histone modifications, can suppress immune-related gene expression, further promoting hyperprogression. For instance, epigenetic silencing of interferon-stimulated genes in *JAK1/2*-mutant tumours has been shown to exacerbate ICI resistance and accelerate tumour progression [137].

By elucidating these molecular mechanisms, ongoing research aims to identify predictive biomarkers and therapeutic strategies to mitigate hyperprogression in lung cancer patients undergoing ICI therapy.

## 4. Strategies to Overcome Resistance to Immunotherapy

Overcoming immunotherapy resistance represents one of the most significant challenges in lung cancer treatment, particularly in non-small cell lung cancer (NSCLC) [129]. As our understanding of resistance mechanisms continues to evolve, several innovative strategies have emerged to address both primary and acquired resistance, considering factors such as comorbidities, genomic alterations, and changes in the tumour microenvironment (TME).

### 4.1. Combination Therapies

The development of combination therapies has marked a significant advancement in overcoming immunotherapy resistance. These approaches combine immune checkpoint inhibitors (ICIs) with various treatment modalities, including chemotherapy, targeted therapies, anti-angiogenic agents, and other modern drugs [164]. The future of lung cancer treatment lies in research on novel immunotherapies, both as monotherapies and in innovative drug combinations [165].

★
**Chemotherapy and Immunotherapy Combinations**


Chemotherapy-immunotherapy combinations have demonstrated particular efficacy through their ability to trigger immunogenic cell death [166]. The landmark KEYNOTE-189 trial provided compelling evidence for this approach, showing significantly improved survival outcomes when pembrolizumab was combined with chemotherapy in NSCLC [44]. Similarly, studies by Reck et al. demonstrated that atezolizumab, when paired with bevacizumab and chemotherapy, improved progression-free survival and overall survival, regardless of PD-L1 expression and EGFR or ALK genetic alteration status [45].

★
**Targeted Therapy Combinations**


In patients with specific molecular alterations, combining targeted therapies with immunotherapy offers a promising strategy, particularly in overcoming resistance associated with long-term targeted therapy use. Studies by Nakagawa et al. have examined the dual blockade of EGFR and VEGF pathways in EGFR-mutated metastatic NSCLC, with phase three trials of EGFR TKI combined with ramucirumab showing positive outcomes on progression-free survival [57].

Combination therapies targeting CTLA-4 (ipilimumab) and PD-1/PD-L1 (nivolumab) have shown improved survival in NSCLC. Ongoing phase 3 trials are investigating coinhibitory targets such as TIGIT, VISTA, LAG3, TIM3, and Siglec-15, where tiragolumab (anti-TIGIT) combined with atezolizumab (anti–PD-L1) has demonstrated increased overall response rate and progression-free survival in PD-L1-positive NSCLC patients [28,102]. Additionally, promising results have emerged from research combining anti-VEGF agents (like bevacizumab or lenvatinib) with ICI [28,45].

★
**Radiation Therapy Combinations**


Radiotherapy, traditionally used for local tumour control, has demonstrated significant potential when combined with immunotherapy. For patients with limited progression sites, this combination has shown potential benefits [167], with studies by Theelen et al. demonstrating improved overall survival in NSCLC patients when radiotherapy was combined with pembrolizumab [52]. The synergistic effect extends beyond local control, with some cases demonstrating the abscopal effect, where non-irradiated metastatic sites also show response to treatment.

★
**Novel Combinations and Emerging Approaches**


Current research is exploring various pathways involving transforming growth factor β (TGFβ), IL-1β, PARP, VEGF, and cMET [28]. Although initial studies suggested that targeting inflammation (and thus cytokines) may be beneficial in diseases such as cancer, research on IL-1β inhibition (canakinumab) as well as other cytokine targets, like IL-10 and TGF-β, did not demonstrate an overall survival benefit in phase III trials which can be related to several factors [28,168]. This is largely due to the complex role of cytokines in the immune system: some cytokines promote tumour growth, while others are critical for immune surveillance [168]. For example, inhibiting IL-1β can reduce inflammation but can also impair antitumour immune responses [28]. Similarly, IL-10 and TGF-γ play roles in both inhibiting and promoting tumour progression, depending on the context [168]. Ultimately, these therapies have failed to demonstrate consistent clinical benefit in late-phase studies [28,168].

The combination of BRAF and mitogen-activated protein kinase (MEK) inhibitors (dabrafenib and trametinib) has shown efficacy in BRAF V600E-mutant NSCLC, as demonstrated by Planchard et al. [112]. Additionally, therapeutic cancer vaccines, when combined with checkpoint inhibitors, aim to potentiate immune responses against cancer [113].

### 4.2. Targeting the Tumour Microenvironment

The tumour microenvironment plays a crucial role in immunotherapy resistance, necessitating targeted strategies to modify this complex ecosystem [169]. Current approaches focus on inhibiting immunosuppressive cells and cytokines, modulating tumour metabolism, and enhancing T-cell trafficking, survival, and function [128]. The phase II COAST trial, evaluating the anti-CD73 antibody oleclumab in combination with durvalumab in stage III NSCLC, represents a significant step forward in targeting the TME [170].

Novel intratumoral therapies, including toll-like receptor agonists, STING agonists, and oncolytic viruses, are being investigated to modulate the TME and enhance ICI efficacy [171]. Additionally, nanomedicine delivery systems are being developed to enhance the local concentration of ICIs and reduce systemic side effects [172].

### 4.3. Addressing Genomic Alterations

The identification of specific genetic mutations driving resistance to immune checkpoint inhibitors (ICIs) has led to the development of targeted therapeutic strategies tailored to distinct molecular profiles. Among these, mutations in STK11, KEAP1, and JAK1/2 have been recognised as key determinants of immune response and treatment outcomes. Current approaches aim to counteract the immunosuppressive effects of these mutations and enhance the efficacy of immunotherapy.

Mutations in STK11 create a non-T cell-inflamed tumour microenvironment (TME), which reduces responsiveness to immune checkpoint inhibitors (ICIs). To overcome this resistance, researchers have explored combination therapies that integrate CTLA-4 inhibitors, such as tremelimumab, to enhance T-cell priming. Additionally, efforts to modulate the inflammatory landscape through IL-6-neutralising or neutrophil-depleting antibodies show promise in reversing the immunosuppressive effects of STK11-driven tumours. Metabolic interventions targeting tumour-associated metabolic reprogramming may further restore immune activity and improve responses to ICIs [140].

Similarly, KEAP1 mutations drive hyperactivation of the NRF2 pathway, promoting immune evasion through metabolic reprogramming. Recent studies highlight the potential of combining glutaminase inhibitors with ICIs to exploit metabolic vulnerabilities in KEAP1-mutant tumours. Other approaches focus on pathway modulation to counteract NRF2-driven immune suppression, offering new avenues for restoring immune sensitivity [136].

In contrast, JAK1/2 mutations disrupt interferon-gamma signalling, impairing antigen presentation and T-cell activation. Strategies aimed at restoring or bypassing interferon signalling pathways have been proposed to enhance immune recognition in these tumours. Therapeutic approaches that improve antigen presentation and bolster T cell activation through combination therapies are also being investigated to counteract this form of resistance [137].

By tailoring treatment strategies to the underlying genetic landscape, these targeted approaches aim to overcome immunotherapy resistance and extend the benefits of ICIs to a broader range of patients. Ongoing research continues to refine these interventions, bringing new hope for more effective and personalised treatments for lung cancer.

### 4.4. Impact of Comorbidities

Comorbidities, particularly chronic obstructive pulmonary disease (COPD), can significantly influence immune responses and treatment outcomes. Optimising COPD treatment has shown the potential to improve overall immune function and potentially enhance the efficacy of ICIs [28]. Management of comorbidities represents an essential component of comprehensive immunotherapy resistance strategies.

### 4.5. Novel Biomarkers and Immune Checkpoints

The exploration of new immune checkpoints and biomarkers has emerged as a crucial strategy for overcoming resistance. PD-L1 expression and tumour mutation burden (TMB) remain the most applied biomarkers for predicting immunotherapy response [173], with NSCLC patients showing high PD-L1 expression typically obtaining better benefits from anti-PD-1/PD-L1 therapy [174].

Recent advances include investigation of novel checkpoint inhibitors targeting LAG-3, TIM-3, and other proteins. Relatlimab, a LAG-3 inhibitor, combined with nivolumab has shown promising activity in PD-L1-positive NSCLC [114]. Similarly, sabatolimab, an anti-TIM-3 IgG4 monoclonal antibody, has demonstrated potential in combination studies [26,116].

TIGIT blockade has shown particular promise, with tiragolumab combination therapy significantly enhancing progression-free survival and overall response rate in PD-L1-positive patients [117]. The development of ICOS (Inducible T-Cell Co-Stimulator) targeting approaches, such as GSK3359609, has also shown encouraging early results [120].

### 4.6. Liquid Biopsy for Detecting ICI Sensitivity and Resistance Biomarkers

Liquid biopsy is an emerging, minimally invasive technique that analyses circulating tumour-derived components in bodily fluids, offering significant potential in oncology for detecting biomarkers associated with sensitivity and resistance to immune checkpoint inhibitors (ICIs). Key analytes in liquid biopsies include circulating tumour DNA (ctDNA), circulating tumour cells (CTCs), and extracellular vesicles, which provide insights into the tumour’s genetic and epigenetic landscape [175,176].

Liquid biopsies can identify biomarkers predictive of a positive response to ICIs. For instance, tumour mutational burden (TMB) can be assessed through ctDNA analysis, providing a non-invasive means to predict ICI efficacy. High TMB levels have been associated with better responses to ICIs across various cancers. Additionally, monitoring dynamic changes in ctDNA levels during treatment can offer real-time insights into therapeutic responses [177].

Liquid biopsies are instrumental in identifying mechanisms of resistance to ICIs. For example, specific genetic alterations detected in ctDNA, such as mutations in the JAK1/2 genes, have been linked to acquired resistance to PD-1 blockade therapies. Furthermore, the detection of CTCs expressing immune checkpoint ligands like PD-L1 can provide insights into tumour immune evasion strategies. The primary advantage of liquid biopsy lies in its ability to provide a comprehensive and dynamic overview of tumour heterogeneity and evolution, which is often challenging with traditional tissue biopsies. However, challenges remain, including standardising methodologies, ensuring analytical sensitivity and specificity, and validating findings across large patient cohorts. Despite these hurdles, liquid biopsy holds promise for guiding immunotherapy decisions and monitoring treatment responses in real time [178,179,180].

### 4.7. Adaptive Clinical Trial Designs

Adaptive clinical trial design can significantly help in overcoming resistance to immunotherapy in lung cancer by allowing for real-time modifications based on interim data, thereby addressing the roles of metabolic, microbial, and epigenetic factors, as well as genetic mutations.

Adaptive trials can incorporate biomarker-driven strategies to tailor treatments to individual patient profiles. For instance, patients with *STK11* mutations, which are associated with an immunosuppressive tumour microenvironment (TME), can be stratified and treated with combination therapies that include immune checkpoint inhibitors (ICIs) and metabolic modulators to counteract the metabolic reprogramming that promotes resistance [181,182].

For *KEAP1* mutations, which lead to hyperactivation of the NRF2 pathway and immune evasion, adaptive trials can evaluate the efficacy of combining ICIs with glutaminase inhibitors. This approach targets the metabolic vulnerabilities of *KEAP1*-mutant tumours, potentially enhancing the response to immunotherapy [181,183].

*JAK1/2* mutations disrupt interferon-gamma signalling, impairing antigen presentation and T cell activation. Adaptive trial designs can facilitate the testing of agents that restore or bypass these pathways, such as epigenetic therapies that enhance antigen presentation or T cell activation, thereby improving outcomes in *JAK1/2*-mutant tumours [139,181].

Additionally, adaptive trials can incorporate microbial factors by monitoring the gut microbiome and adjusting treatments based on microbiome composition, which has been shown to influence the efficacy of ICIs [181,182].

By using master protocols and biomarker-adaptive randomization, adaptive trials can efficiently identify and validate the most effective treatment combinations for overcoming resistance, ultimately improving overall survival (OS) and progression-free survival (PFS) in lung cancer patients [184,185,186].

### 4.8. Recent Clinical Trials Targeting Resistance to Immunotherapy in Lung Cancer

Several recent clinical trials are targeting resistance to immunotherapy in lung cancer, particularly for patients with non-small cell lung cancer (NSCLC) and small cell lung cancer (SCLC), considering factors such as comorbidities like chronic obstructive pulmonary disease (COPD), genomic alterations, and changes in the tumour microenvironment (TME). They are summarised in Table 5.

## 5. Future Directions

Despite recent progress in precision medicine and numerous ongoing clinical trials, several critical challenges and questions remain to be addressed, namely:

### 5.1. Treatment Response and Resistance

First of all, immunotherapy response rates still remain relatively low. Current combination therapies, while more effective than single agents, still achieve response rates below 50%. PD1/L1 expression alone has proven insufficient as a predictor of response. Even among responding patients, complete tumour elimination is rare, even if occurs, especially in bladder and melanoma cancer patients. Remaining cells may lead to relapse, often developing different immunosuppression mechanisms and mutations compared to the original tumour.

Probably treatment optimization might be the key, apart from other factors, such as biomarkers. The optimal combination and sequencing of targeted therapy, radiotherapy, chemotherapy, and immunotherapy remain unclear, despite many ongoing trials. Questions persist about the timing of immunotherapy, particularly regarding the patient’s immune system status after chemotherapy. Finally, the role of neoadjuvant chemotherapy requires further investigation.

### 5.2. Biological Understanding and Patient Factors

Our understanding of cancer biology remains in its infancy. The durability of responses to immunotherapy likely connects to biological mechanisms that are yet to be discovered, requiring further basic science research to understand the complex pathways involved. Moreover, adverse effects and the hyperprogression phenomenon also require more attention, particularly regarding the variability in adverse effect severity among patients. Some patients develop aggressive autoimmune progression or hyperprogression despite initially having very good prognostic biomarkers. Factors involved may include immune system function, organism-wide effects, or unidentified environmental influences.

Finally, regarding gender and racial differences: research has revealed significant variations between males and females in gene expression and methylation status, suggesting the need for gender-specific treatment approaches. This issue has been recently raised by many international committees and, as a result, more clinical trials will encompass inclusiveness or will be conducted separately for specific ethnic groups.

### 5.3. Future Research Priorities

The path forward requires a coordinated effort from the scientific community, combining basic research with clinical applications. Integration of multi-omics approaches, and advanced modelling techniques will be essential for unravelling the complexities of cancer biology and improving patient outcomes. Success will depend on developing personalised, multi-modal approaches tailored to each patient’s unique tumour and immune profile, supported by adaptive trial designs and biomarker-driven treatment selection.

The most important research directions for lung immunotherapy have been presented in bullet points below:
Biomarker Development:
urgent need for reliable biomarker detection,potential of liquid biopsy techniques using circulating tumour DNA,development of markers for tumour presence, progression, metastases, and treatment response,identification of markers to predict treatment benefits and adverse effect risks.Treatment Optimization:
precise dosage administration,expansion of pharmacogenetics and pharmacogenomics,implementation of genetic testing before drug administration,standardization of resistance definitions,development of comprehensive genomic profiling with immune gene signatures.Novel Therapeutic Approaches:
personalised cancer vaccine development,investigation of epigenetic modulation,metabolic manipulation strategies,gut microbiome modulation,exploration of immunotherapy rechallenge in previously responsive patients.Neoantigen Research:
further investigation into the stochastic nature of neoantigen expression and its targeting potential.


### 5.4. Future Directions and Remaining Open Questions

In summary, despite recent advances in precision medicine and numerous ongoing clinical trials, several critical questions remain unresolved. We propose addressing these challenges collectively as a scientific community:The limited response rate to immunotherapy remains problematic. Although combination therapies demonstrate enhanced efficacy, response rates rarely exceed 50% and are typically substantially lower. PD-1/PD-L1 expression alone proves insufficient as a predictive biomarker for potential responders.Even among responsive patients, complete tumour cell elimination is rare, with residual cells potentially facilitating future relapse through the development of resistant clones. Such recurrent tumours frequently exhibit altered immunosuppressive mechanisms and mutational profiles distinct from their progenitor lesions. Consequently, the optimization of first-line therapy, specifically tailored to individual patients, remains crucial.The optimal integration of targeted therapy, radiotherapy, chemotherapy, and/or immunotherapy remains undefined. Similarly, the appropriate therapeutic sequence requires clarification, as preliminary data suggest immunotherapy efficacy may be compromised in patients who have undergone chemotherapy due to immune system exhaustion or suppression. The role of neoadjuvant chemotherapy, in particular, warrants further investigation.When effective, immunotherapeutic responses demonstrate remarkable durability. However, the biological mechanisms underlying this sustained response remain poorly elucidated and necessitate fundamental scientific investigation. Enhanced comprehension of cancer biology will facilitate more precise targeting of critical nodes within complex pathways.The heterogeneity in adverse event severity among patients receiving immunotherapy remains unexplained. This variability may be attributable to intrinsic immune system function, systemic physiological factors, or unidentified environmental variables that have yet to be characterised.The etiology of hyperprogressions and severe autoimmune complications following immunotherapy administration remains unclear. A subset of patients experiences rapid deterioration and mortality shortly after immune checkpoint inhibitor initiation. Current knowledge is insufficient to explain this phenomenon adequately or to implement appropriate interventional strategies.While it is now widely accepted that tumours express targetable neoantigens, their expression patterns appear stochastic, necessitating substantial additional research in this domain.Therapeutic dosage optimization requires precision, yet sufficient data to accomplish this effectively remains lacking. Pharmacogenetics and pharmacogenomics offer promising approaches, though these analyses are not universally implemented. In many regions, they remain insufficiently adopted or are employed reactively following adverse events rather than proactively before treatment initiation.There exists an urgent requirement for robust biomarker identification. Emerging diagnostic technologies based on circulating tumour DNA (ctDNA)—termed liquid biopsy—hold significant potential for identifying molecules that serve as biomarkers for tumour presence, progression, metastasis, treatment selection, and therapeutic response monitoring. Specific biomarkers would also facilitate the identification of patients most likely to benefit from therapy while minimising adverse effects.Multiple investigations have demonstrated significant sex-based differences in gene expression and methylation status. These findings suggest that therapeutic approaches should be further refined according to patients’ biological sex among other relevant factors. In oncology, precision medicine represents a necessity rather than an optional approach for clinicians or institutions.Although numerous factors are known to influence patient response to immunotherapy, our limited understanding of underlying mechanisms limits their clinical application. For instance, low-light conditions are known to induce a more inflammatory microenvironment, potentially enhancing immunotherapeutic efficacy while possibly compromising mental health. Similar considerations apply to caloric restriction.

### 5.5. Insights from the Genomic Medicine Perspective

Genetic variants may influence the prediction of the treatment of lung cancer. For example, a set of 140 genes may predict the response and survival to anti-PD-L1 therapy in non-small cell lung cancer. The set of genes may help to identify 5 classes of clinically aggressive tumours that can respond to immunotherapy. In turn, in the case of lung adenocarcinoma, a set of 10 genes and their genetic variants associated with chemotherapy resistance has been identified (namely, PLEK2, TFAP2A, KIF20A, S100P, GDF15, HSPB8, SASH1, WASF3, LAMA3 and TCN1). Further, a subset of genes related to a particular clinical outcome may be delineated. For example, TP53 alterations and co-alterations of TP53 missense mutations with TP73, CREBBP/EP300 or FMN2 are indicative of a shorter disease prolapse after chemotherapy and some genetic patterns associated with resistance to chemotherapy [193]. The most important genetic insights have been summarised in Table 6. Furthermore, deep learning algorithms may identify expression profiles related to the metastatic potential of the tumour [194].

## 6. Conclusions

Lung cancer remains an example of malignancy with a very poor prognosis despite the recent progress in medicine. Different treatment strategies are now available for cancer therapy based on its type, molecular subtype and other factors including overall health, the stage of disease and cancer molecular profile. Immunotherapy is emerging as a potential addition to surgery, chemotherapy, radiotherapy or other targeted therapies, but also considered a mainstay therapy mode. In fact, the pace of research and clinical studies on cancer immunotherapy seem to outride the progress in understanding the biological background underlying behind. The lung cancer microenvironment is a complex and dynamic entity that significantly influences tumour behaviour and therapeutic response. Continued research into the cellular and molecular mechanisms within the TME will provide insights into novel therapeutic targets and strategies. Integrating multi-omics approaches and advanced modelling techniques will be essential for unravelling the intricacies of the TME and improving outcomes for lung cancer patients [195,196].

## Figures and Tables

**Figure 1 cells-14-00587-f001:**
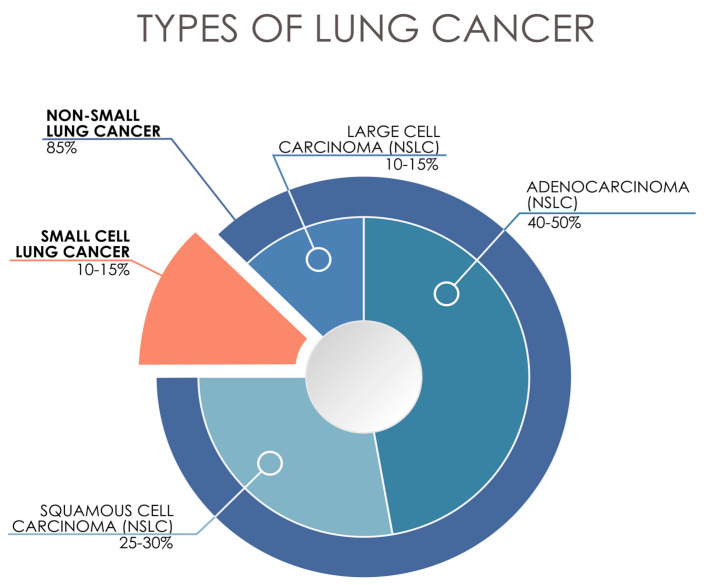
Typology of lung cancer currently used in the clinic, based on histological features.

**Figure 2 cells-14-00587-f002:**
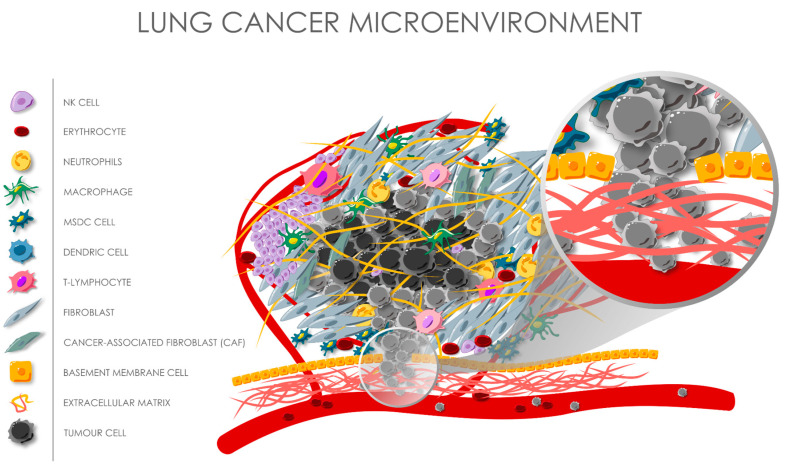
Microenvironment of the lung tumour, depicting the level of cancer heterogeneity, complicated intricacies between different cellular and noncellular components, intratumoural hypoxia regions, as well as the possibility of further metastasis due to the free tumour circulating cells crossing the basement membrane into the bloodstream.

**Table 3 cells-14-00587-t003:** Results of Checkpoint Inhibitor Trials Targeting LAG-3, TIM-3, and Other Emerging Pathways.

Target	Drug	Trial Phase	Population	Key Findings
LAG-3	Relatlimab	Phase I/II	Advanced NSCLC	Combined with nivolumab: improved progression-free survival (PFS) in PD-L1-positive tumours. Median PFS: 6.4 months.
LAG-3	Eftilagimod alpha (IMP321)	Phase II	NSCLC, first-line	Combined with pembrolizumab: 47% disease control rate (DCR), well-tolerated.
TIM-3	Sabatolimab (MBG453)	Phase I	Advanced solid tumours, including NSCLC	Early safety and efficacy data suggest manageable toxicity; limited antitumour activity as monotherapy.
TIM-3	TSR-022	Phase I/II	NSCLC	Combined with anti-PD-1: objective response rate (ORR) 25% in heavily pretreated patients.
TIGIT	Tiragolumab	Phase II	NSCLC, PD-L1-positive	Combined with atezolizumab: PFS benefit (5.6 months vs. 3.9 months) compared to atezolizumab alone.
TIGIT	Domvanalimab	Phase II	Advanced NSCLC	Combined with zimberelimab: ORR of 27.3%, encouraging safety profile.
ICOS	GSK3359609	Phase I	Advanced solid tumours, including NSCLC	Partial responses observed; limited monotherapy efficacy but potential in combinations.

**Table 4 cells-14-00587-t004:** Metabolic changes and therapeutic strategies of STK11 (LKB1), KEAP1, and JAK1/2 Mutations.

Mutation	Metabolic Changes	Therapeutic Strategies
*STK11* *(LKB1)*	epigenetic changes (methylation, histone acetylation) increased lactate production M2 macrophage polarization and T cell dysfunction	DNA methyltransferase inhibitors (DNMTis) or histone deacetylase inhibitors (HDACis)
*KEAP1*	hyperactivation of the NRF2 pathway increasing the consumption of glucose and glutamine by tumour cells T cell deprived of nutrients	combining ICIs with glutaminase inhibitors epigenetic regulation of the NRF2 pathway
*JAK1/2*	disrupt interferon-gamma signalling	agents that enhance antigen presentation or T cell activation

**Table 5 cells-14-00587-t005:** Summary of recent clinical trials targeting resistance to immunotherapy in lung cancer.

Trial Name and Reference	Cancer Type	Investigated Treatment Regimen	Phase	Study Population	Results	Overcoming Resistance Strategy
CheckMate 227 [157]	NSCLC	Nivolumab + Ipilimumab vs. Chemotherapy	III	Patients with Stage IV or recurrent NSCLC who had not received prior systemic therapy	Significantly improved overall survival on combination of nivolumab and ipilimumab compared to chemotherapy alone	Multiple immune checkpoints
KEYNOTE-598 [187]	NSCLC	Pembrolizumab + Ipilimumab vs. Pembrolizumab	III	Patients with previously untreated metastatic NSCLC with a PD-L1 tumour proportion score (TPS) ≥ 50% and no sensitising EGFR or ALK aberrations	Lack of improved efficacy and greater toxicity of combination therapy	Multiple immune checkpoints
ADRIATIC [188]	SCLC	ICI maintenance therapy—Durvalumab (+Tremelimumab)	III	Patients with limited-stage small-cell lung cancer (LS-SCLC) who have not progressed after concurrent chemoradiotherapy (cCRT)	Trial is ongoing	Multiple immune checkpoints
IMpower133 [189]	SCLC	Atezolizumab + Carboplatin + Etoposide vs. Carboplatin + Etoposide	III	Patients with previously untreated extensive-stage small-cell (ES-SCLC) lung cancer	Combination of atezolizumab with carboplatin and etoposide as a new standard of care for first-line treatment of ES-SCLC.	Enhanced immune response
STIMULI [190]	SCLC	Consolidation Immunotherapy with Nivolumab + Ipilimumab	II	Patients with limited-stage small-cell lung cancer (LS-SCLC) who had not progressed after concurrent chemoradiotherapy (cCRT) and prophylactic cranial irradiation (PCI)	The trial did not meet its primary endpoint of improving progression-free survival (PFS)	Multiple immune checkpoints
CONTACT-01 [191]	NSCLC	Atezolizumab + Cabozantinib vs. Docetaxel	III	Patients with metastatic non-small cell lung cancer (NSCLC) who had progressed after prior treatment with a checkpoint inhibitor and platinum-containing chemotherapy	Not significantly improved OS but significantly improved PFS	Combining immune checkpoint inhibition with the anti-angiogenic and immunomodulatory effects of cabozatinib
SAPPHIRE [192]	NSCLC	Sitravatinib **+** Nivolumab vs. Docetaxel	III	Patients with advanced nonsquamous non-small cell lung cancer (NSCLC) who had progressed after prior treatment with a checkpoint inhibitor and platinum-containing chemotherapy	No significant improvement of OS or PFS in Sitravatinib + Nivolumab group compared to Docetaxel	Combining immune checkpoint inhibition with the with immunomodulatory effects of sitravatinib

**Table 6 cells-14-00587-t006:** Key points regarding genomic medicine in lung cancer—examples from the current landscape and future directions [52,67,70,77,110].

***EGFR* mutations**	Patients with activating EGFR mutations (e.g., exon 19 deletion, L858R) respond well to EGFR TKIs like **Gefitinib** and **Erlotinib**. Mutations like **T790M** drive resistance, but **Osimertinib** is effective against this mutation.
***ALK* rearrangements**	Found in about 3–7% of NSCLC, these rearrangements respond to ALK inhibitors like **Alectinib** and **Crizotinib**. Newer agents like **Lorlatinib** work against ALK inhibitor-resistant mutations.
***KRAS* mutations**	The ***KRAS* G12C** mutation was historically undruggable, but **Sotorasib** and **Adagrasib** are now effective therapies for this subset.
***MET* exon 14 skipping**	Therapies like **Capmatinib** and **Tepotinib** target this mutation, which occurs in about 3–4% of NSCLC.
***RET* and *ROS1* fusions**	Drugs such as **Selpercatinib** and **Crizotinib** target fusions in these genes. **Entrectinib** also targets ***NTRK*** gene fusions, which occur less frequently.
**PD-1/PD-L1 inhibitors**	Immunotherapies like **Pembrolizumab**, **Nivolumab**, and **Atezolizumab** are approved based on high PD-L1 expression and have become standard treatments for NSCLC.

## Data Availability

This article is licensed under a Creative Commons Attribution 4.0 International License, which permits use, sharing, adaptation, distribution and reproduction in any medium or format, as long as you give appropriate credit to the original author(s) and the source, provide a link to the Creative Commons licence, and indicate if changes were made.

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
