# Peer review of "Resistance in Lung Cancer Immunotherapy and How to Overcome It: Insights from the Genetics Perspective and Combination Therapies Approach"

_cells, 2025, doi:10.3390/cells14080587_

Round 1

Reviewer 1 Report (New Reviewer)

Comments and Suggestions for Authors

In this review manuscript, the authors provide information about immunotherapy, especially immune checkpoint inhibitors (ICIs) and how this treatment is helpful for the immune system to recognize and destroy cancer cells. Whereas resistance towards immunotherapy is a major challenge. This resistance process includes genetic alterations (e.g., STK11, KEAP1, JAK1/2 mutations), metabolic changes, microbiome influence, and the immunosuppressive tumor microenvironment. Hyperprogression, the process of rapid growth in tumor cells is also linked with these metabolic alterations processes. Overall, I recommend this paper for publication in Cells journal as it effectively addresses important issues related to lung cancer. However, it would be beneficial to update a few aspects. Comments In section 3.2.1 the STK11, KEAP1, and JAK1/2 mutations are discussed multiple times it will be better to summarize it in a table for better understanding. In 3.2.4, the immunotherapy response towards COPD is unclear either it enhances or reduce. More comprehensive review literature should be provided to improve clarity. Add additional studies that explore the mechanism and therapeutic effect of immunotherapy in COPD patients. In 483,484 it is discussed that IL-1β inhibitor (canakinumab), and other cytokine targeting therapies fail in Phase III. A brief explanation is required so that the readers can understand the reasons behind these failures. Some of the references are repeated across various sections leading to redundancy in the literature review. It would be better to find any other literature related to the topic and use those references to offer more comprehensive and thorough discussion. In section 5.2 the first line lack clarity and should be rephrased to make it more precise and understandable for the readers. In section 5.4 there are some grammatical mistakes that need correction and the sentence “If they do work, indeed, the response is rather durable…” needs to be rewrite in a more formal and academic tone.

Author Response

Dear Reviewer,

Thank You very much for Your time and effort, as well as constructive comments concerning our manuscript. We have studied Your comments carefully, had team discussions and made few significant corrections and improvements, which we hope to meet with Your approval. Thanks to this work being done, we were also able to expand our knowledge even more - and enhance our future projects.

  • Comments In section 3.2.1 the STK11, KEAP1, and JAK1/2 mutations are discussed multiple times it will be better to summarize it in a table for better understanding.

We thank the Reviewer for this comment and gathered the information regarding STK11, KEAP1, and JAK1/2 mutations in the table.

  • In 3.2.4, the immunotherapy response towards COPD is unclear either it enhances or reduce. More comprehensive review literature should be provided to improve clarity. Add additional studies that explore the mechanism and therapeutic effect of immunotherapy in COPD patients.

This has been done, thank you for your comment.

  • In 483,484 it is discussed that IL-1β inhibitor (canakinumab), and other cytokine targeting therapies fail in Phase III. A brief explanation is required so that the readers can understand the reasons behind these failures.

This part has been changed to explain the factors influencing cytokine targeting therapies failures.

  • Some of the references are repeated across various sections leading to redundancy in the literature review. It would be better to find any other literature related to the topic and use those references to offer more comprehensive and thorough discussion.

The references section has been changed, thank you.

  • In section 5.2 the first line lack clarity and should be rephrased to make it more precise and understandable for the readers.

Section 5.2 has been slightly rephrased to improve the clarity.

  • In section 5.4 there are some grammatical mistakes that need correction and the sentence “If they do work, indeed, the response is rather durable…” needs to be rewrite in a more formal and academic tone.

Section 5.4 has been rephrased to have a more academic tone.

We decided to highlight all added or changed paragraphs (yellow highlighter) to facilitate the process of reviewing and save Your time.

We really hope these modifications can meet with Your approval.

Thank You very much for Your time and effort. We know that nowadays time is scarce. 

Yours Sincerely,

Authors

Reviewer 2 Report (New Reviewer)

Comments and Suggestions for Authors

The manuscript is timely relevant and technically correct requiring moderate suggestions to be accepted for the publication on this journal

  • In the introduction section, please, could the authors concisely assess the gap and the advantages in LC patients?
  •  In the manuscript, please, could the authors provide more details on first line and second line ICs approaches showing the most relevant clinical aspects?
  • In the manuscript, please, could the authors check if figure 2 has been properly generated?
  • Please, could the authors review table 1 improving readability on this journal?
  • Please, could the authors also add some details on the role of liquid biopsy both to detect sensitive and resistance ICIs biomarkers?
  • extensive native english revision should be approached to improve acceptability on this journal
Comments on the Quality of English Language
  • extensive native english revision should be approached to improve acceptability on this journal

Author Response

Dear Reviewer,

Thank You very much for Your time and effort, as well as constructive comments concerning our manuscript. We have studied Your comments carefully, had team discussions, and made few significant corrections and improvements, which we hope to meet with Your approval. Thanks to this work being done, we were also able to expand our knowledge even more - and enhance our future projects.

  • In the introduction section, please, could the authors concisely assess the gap and the advantages in LC patients?

This has been done, thank you for your comment.

  • In the manuscript, please, could the authors provide more details on first line and second line ICs approaches showing the most relevant clinical aspects?

This has been done, thank you for your comment.

  • In the manuscript, please, could the authors check if figure 2 has been properly generated?

Figure 2 is a conceptual diagram based on numerous publications on the tumour microenvironment. It illustrates the key cellular components of the immune system and the anatomical structures essential for understanding the interactions within the tumour tissue. We agree, however, that the tumour microenvironement is a very unstable phenomenon and can be very different among patients, as well as different stages of the disease. Therefore, this picture shows the most frequently described lung cancer TME composition.

  • Please, could the authors review table 1 improving readability on this journal?

Table 1 has been edited, however, to better fit the journal editorial style it will be further the journal’s editorial team, in order to improve its readability and adjust it to the journal rules.

  • Please, could the authors also add some details on the role of liquid biopsy both to detect sensitive and resistance ICIs biomarkers?

Very good idea, we’ve done it with pleasure, thank you for your comment.

  • extensive native english revision should be approached to improve acceptability on this journal

Thank you, we’ve done this too.

We decided to highlight all added or changed paragraphs (yellow highlighter; except for the language changes) to facilitate the process of reviewing and save Your time.

We really hope these modifications can meet with Your approval.

Thank You very much for Your time and effort. We know that nowadays time is scarce. 

Yours Sincerely,

Authors

Round 2

Reviewer 2 Report (New Reviewer)

Comments and Suggestions for Authors

no other comments

This manuscript is a resubmission of an earlier submission. The following is a list of the peer review reports and author responses from that submission.

Round 1

Reviewer 1 Report

Comments and Suggestions for Authors

The title of the manuscript is "How to Overcome Immunotherapy Resistance in Lung Cancer?" The content did not concentrate on "Insights from the Genetics Perspective and Combination Therapies Approach." The overall structure is bizarre and convoluted with the current treatment rationale. The content also did not address immunotherapy resistance. The author should be more concise and align with the title. 

Comments on the Quality of English Language

average 

Reviewer 2 Report

Comments and Suggestions for Authors

The manuscript does not align well with its title, which emphasizes immunotherapy resistance. While the title suggests a focus on this critical topic, the majority of the manuscript discusses treatments unrelated to immunotherapy, such as surgery, chemo, radio and targeted therapies (from page 2 to 13). An additional small portion is devoted to the tumor microenvironment (from page 13 to 15) , with extremely small section of the content addressing immunotherapy specifically (from page 15 to 17).

None of the tables or figures are relevant to immunotherapy response/resistance. Furthermore, the authors have failed to comprehensively review the current findings key aspects of immunotherapy, such as immune checkpoint inhibitors (ICIs), existing molecular biomarkers like PD-L1 expression, tumor mutation burden, immunoscore, and neoantigens, as well as their roles as predictive markers for treatment response and resistance. The manuscript also lacks in-depth discussion of the current challenges and future directions in understanding and overcoming immunotherapy resistance, which are central to the topic. 

Overall, the manuscript is unfocused and does not match the title, and did not bring the content in line with the stated topic.